# Evaluation of Urban Traffic Accidents Based on Pedestrian Landing Injury Risks

**Liangliang Shi** [1] , **Ming Liu** [1] **, Yu Liu** [1] **, Qingjiang Zhao** [1] **, Kuo Cheng** [1] **, Honghao Zhang** [1,2,] * 
**and Amir M. Fathollahi-Fard** [3]

[1]   State Key Laboratory of Vehicle NVH and Safety Technology, China Automotive Engineering Research
     Institute Co., Ltd., Chongqing 401122, China; shiliangliang@caeri.com.cn (L.S.); liuming@caeri.com.cn (M.L.);
     ly_wen@hnu.edu.cn (Y.L.); zhaoqingjiang@caeri.com.cn (Q.Z.); chengkuo@caeri.com.cn (K.C.)
[2]   School of Mechanical Engineering, Shandong University, Jinan 250100, China
[3]   Department of Electrical Engineering, École de Technologie Supérieure, University of Quebec,
     Montreal, QC H3C 3P8, Canada; amirmohammad.fathollahifard.1@ens.etsmtl.ca
*   Correspondence: honghao_zhang@sdu.edu.cn

**Abstract:** In comparison with vehicle-to-pedestrian collision, pedestrian-to-ground contact usually results in more unpredictable injuries (e.g., intracranial, neck, and abdominal injuries). Although there are many studies for different applications of such methods, this paper conducts an in-depth analysis of urban traffic pedestrian accidents. The effects of pedestrian rotation angle (PRA) and pedestrian facing orientation (PFO) on head and neck injury risk in a ground contact are investigated by the finite element numerical models and different probabilistic analyses. It goes without saying that this study provides a theoretical basis for the prediction and protection study of pedestrian ground contact injury risk. In our experiments, 24 pedestrian-to-ground simulations are carried out by the THUMS v4.0.2 model considering eight PRAs (0°, 45°, 90°, 135°, 180°, 225°, 270°, 315°, 360°) and three PFOs (x+, x−, y+). Each test was simulated with loading the average linear and rotational velocities that obtained from real-world pedestrian accidents at the pedestrian's center of gravity. The results show that both PRAs and PFOs have significant impacts on head and neck injuries. Head HIC value caused by PRA 0–135° is much higher than that caused by PRA 180–315°. Neck injury risk caused by PRA 180° is the greatest one in comparison with other PRAs. The PRAs 90° and 270° usually induce a relatively lower neck injury risk. For PFO, the risk of head and neck injury was lower than PFOy+ and PFOx+ or PFOx−, which means PFOy+ was a safer landing orientation for both head and neck. The potential risk of head and neck injuries caused by the ground contact was strongly associated with the symmetry/asymmetric features of human anatomy.

**Keywords:** urban traffic accidents; smart cities; ground contact; injury risk; data analysis

## 1. Introduction

The World Health Organization (WHO) has released the Global Status Report on Road Safety, which shows that approximately 1.35 million people die annually from road traffic accidents. More than half of all deaths are among those with the least protection: pedestrians (23%), cyclists (3%), and motorcyclists (28%) [1]. Recent trends in traffic safety research show that scientists and industry are making tremendous efforts to find solutions to improve the safety and performance of vehicle systems [2–4]. For decades, scholars have focused on road safety research in vehicle-to-pedestrian collisions, and a variety of vehicle safety protection devices and vehicle optimization designs have been proposed [5–13]. However, it should be noted that the injuries suffered by pedestrians in road traffic do not only come from vehicles, but also from the ground.

The real-world accident data analysis has shown that ground contact plays a very important role in the cause of pedestrian head injuries [14–16]. While there are still few studies on pedestrian-to-ground contact [17–27]. The main reason is that it is difficult for

accident investigators to obtain the impact traces between pedestrians and the ground, compared to pedestrian-to-vehicle contact. In addition, the injury risk from ground was correlated to a lesser extent with vehicle velocity [17,21], and the ground usually results in more unpredictable injuries [28].

In recent years, some researchers have studied the kinematics of pedestrian landing and have come up with several typical types of pedestrian landing kinematics [17,19,25,29], this provided an understanding for pedestrian-to-ground contact; however, they did not deeply analyze the influence mechanism of landing kinematics on landing injuries. In 2018, Shi et al. [21] proposed the concept of the pedestrian rotation angle ranges (PRARs) to illustrate the relationship between pedestrian landing kinematics and landing injuries, which demonstrated that any landing kinematics could be classified as one of the four PRARs, and the PRARs are highly correlated with the ground contact mechanism and head injury risk. However, pedestrian landing kinematics and injury risk are not only affected by pedestrian rotation angle, but also by a variety of other factors, such as the pedestrian facing orientations and the landing sequence of body parts, which they did not analyze. Further study found that for the same pedestrian rotation angle, there were significant differences in the risk of head injury during landing for different pedestrian facing orientations [30]. However, they did not delve into the mechanism between this factor and pedestrian injury risk.

Having a look at the literature review, it was found that the biomechanical mechanisms of pedestrian injury is closely related to the symmetric/asymmetric phenomena (structures, shapes, morphologies, geometry, direction, models, aesthetics, etc.) in life sciences [31,32]. In response to this intriguing problem, this paper systematically and quantitatively investigated the effect of human anatomy features on body landing injury risk by considering two key kinematic parameters, pedestrian rotation angle (PRA) and pedestrian facing orientation (PFO). To this end, six real-world pedestrian accidents were reconstructed to derive the boundary parameters at the moment of body-to-ground impact. Then, a total of 24 pedestrian landing simulations were carried out using the THUMS v4.0.2 model with considering eight PRAs and three PFOs. The kinematic-based injury criteria such as the head injury criterion (HIC) and the maximum rotation angle of T1 relative to the head were included in the analysis to investigate head and neck injury risk. Moreover, the influence mechanism of human anatomical symmetry/asymmetry on the risk of landing injuries was deeply analyzed. The results of this study are beneficial to provide theoretical guidance for the research on pedestrian injury control strategies in the automotive industry and traffic management departments.

The rest of this paper is summarized as follows: Section 2 establishes the proposed methodology for the evaluation of urban traffic accidents. Section 3 defines our computational results. Section 4 talks about our main insights and significant results from our experiments. Finally, Section 5 provides a summary of this paper with findings, limitations, and future research recommendations.

## 2. Methodology

A set of pedestrian-to-ground impact simulations was implemented with the THUMS v4.0.2 model. Each simulation initiated with the pedestrian in a different PRA and PFO just above the ground. PRA is defined by Shi et al. [21], which was used in this paper to explore the effect of asymmetric anatomical features divided by the transverse plane on the risk of ground contact injuries. PFO is defined as the initial facing orientation of pedestrians in the event of vehicle-to-pedestrian collision, aiming to explore the influence of symmetrical and asymmetrical anatomical features divided by sagittal and coronal planes on ground contact injuries.

### 2.1. Parametric Research Proposal

Figure 1 shows the extraction and loading process of boundary parameters for the pedestrian-to-ground impact simulations. First, six real-world pedestrian traffic accidents

were completely reconstructed with reference to the accident reconstruction method proposed by Shi et al. [16]. The detailed workflow of accident reconstruction is depicted in Figure 2. The key parameters related to accidents with vehicles, pedestrians, roads, and the environment are summarized in Table 1. The reconstructed pedestrian kinematics were compared with the video recordings, as shown in Figure A1 (Appendix A). The quantitative validation of reconstruction results and the linear and angular velocity components at the COG of the pedestrian's body at the moment of 2 milliseconds before the body-to-ground contact were extracted and summarized in Table 2. The average linear velocities along the $x$, $y$, and $z$ axes were calculated to be 2.4 m/s, $-1.0$ m/s and $-5.6$ m/s respectively, and the average angular velocity around the $y$-axis was 4.7 rad/s$^2$. Second, the average linear velocity and angular velocity components were loaded into the COG of THUMS v4.0.2 for body-to-ground collisions. To explore the effect of PFO on body-to-ground contact injury, three PFOs were formed by rotating the initial orientation of a standing pedestrian around the $z$-axis at 90-degree intervals, coding the PFOx+, PFOx−, and PFOy+, respectively. Third, each PFO is rotated around the $y$-axis in 45° increments, resulting in eight different PRAs. Finally, a total of 24 pedestrian-to-ground impact configurations were defined.

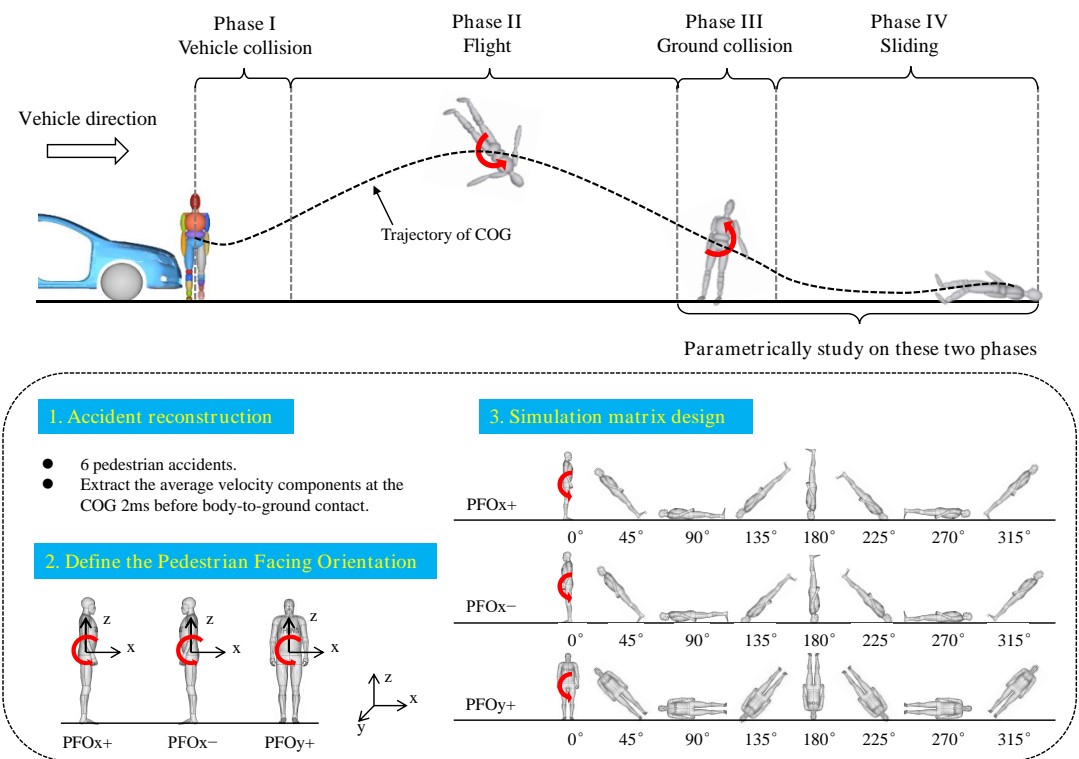

**Figure 1.** Methodology implemented for the parametric study.

## 2.2. Impact Conditions

The ground surface was modeled utilizing a rigid body for representing the asphalt ground [33]. The friction coefficient between pedestrian and ground was set to 0.58 [34]. All numerical simulations were performed using the LS-DYNA MPP R9.3.0, LSTC (Livermore Software Technology Co., Livermore, CA, USA).

## 2.3. Injury Criteria

In this study, the potential risk of head and neck injuries caused by the ground contact was assessed using the HIC and the maximum rotation angle of T1 relative to the head, respectively. The head criteria were selected based on [16], which shown that HIC has a 'good' assessment capability for severe head injuries caused by ground contact. The criteria

value were calculated based on the simulation results using the THUMS v4.0.2 model in the LS-DYNA code.

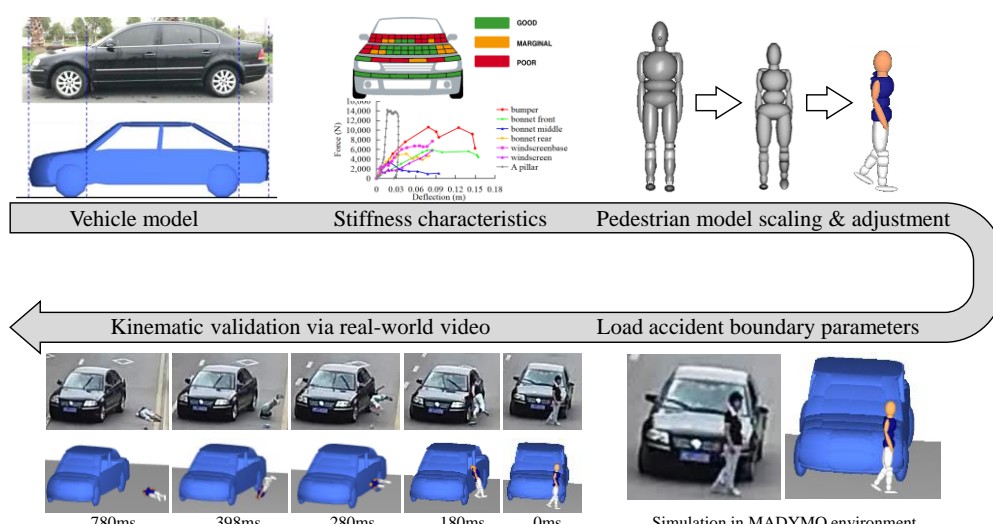

**Figure 2.** Workflow implemented for the pedestrian accident reconstruction.

**Table 1.** Key parameters related to these accidents.

| | Vehicle Info | | | Pedestrian Info | | | | | Road and Environment Info | | | |
|---|---|---|---|---|---|---|---|---|---|---|---|---|
| **Case ID** | **Type** | **Velocity (km/h)** | **Brake before Collision** | **Gender** | **Age** | **Height (cm)** | **State** | **Location** | **Obstacle** | **Pavement** | **Humidity** |
| 1 | Sedan | 67 | No | Male | 36 | 179 | Running | Motor lane | None | Asphalt | Wet |
| 2 | Sedan | 90 | No | Female | 51 | 162 | Running | Motor lane | None | Asphalt | Wet |
| 3 | Sedan | 63 | Yes | Female | 37 | 164 | Running | Motor lane | None | Asphalt | Dry |
| 4 | Sedan | 29 | No | Female | 73 | 142 | Walking | Motor lane | None | Asphalt | Dry |
| 5 | E-bike | 31 | Yes | Female | 63 | 157 | Walking | Non-motor lane | None | Asphalt | Dry |
| 6 | MPV | 29 | No | Female | 79 | 158 | Walking | Crosswalk | None | Asphalt | Dry |

**Table 2.** Validation and extraction of reconstruction results.

| **Case ID** | **Final Position Error %: Reconstructed vs. Actual** | | **Linear and Angular Velocity Components (2 ms before Body-to-Ground Contact)** | | | |
|---|---|---|---|---|---|---|
| | **Pedestrian** | **Vehicle** | **Vx (m/s)** | **Vy (m/s)** | **Vz (m/s)** | **ωy (rad/s)** |
| 1 | 3.1% | 2.4% | 2.8 | −0.4 | −5.8 | 4.6 |
| 2 | 3.6% | 2.7% | 4.7 | 0.1 | −6.9 | 5.8 |
| 3 | 2.4% | 1.7% | 3.9 | 0.2 | −6.1 | 5.2 |
| 4 | 0.9% | 0.6% | 1.3 | −3.8 | −5.5 | 6.5 |
| 5 | 1.2% | 1.1% | 0.9 | −0.8 | −5.2 | 3.8 |
| 6 | 0.6% | 1.3% | 0.6 | −1.4 | −4.3 | 2.4 |

## 3. Results

### 3.1. Pedestrian Landing Kinematics

For the 24 simulations, pedestrian landing kinematics were completely different due to univariate differences in the PRA and PFO settings. Figures 3–5 show three typical kinematic poses for each simulation in the process of body-to-ground contact. Notably, PRA mainly causes changes in the sequence of body parts and ground contact; while PFO mainly caused changes in specific positions of body parts in contact with the ground. For PRAs 0°, 45°, 90°, and 135°, the pedestrian's upper body suffers a linear velocity component (Vz) perpendicular to the ground and an angular velocity component that rotates toward the ground. The superposition of the two velocity components increases the severity of the head-to-ground contact. Whereas for PRAs 180°, 225°, 270°, and 315°, the angular velocity

component causes the head to rotate away from the ground, which can effectively reduce the severity of head-to-ground contact.

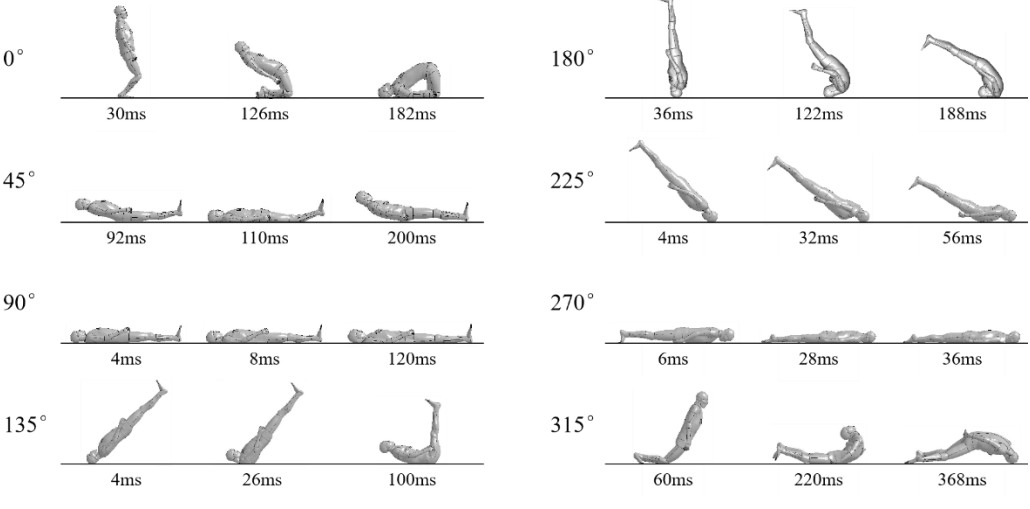

**Figure 3.** Typical landing kinematics in PFOx+.

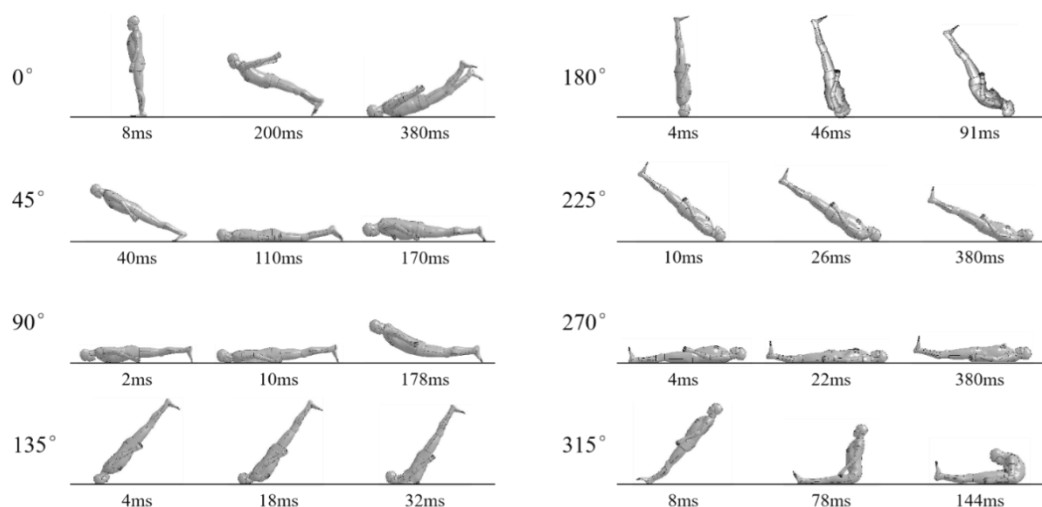

**Figure 4.** Typical landing kinematics in PFOx−.

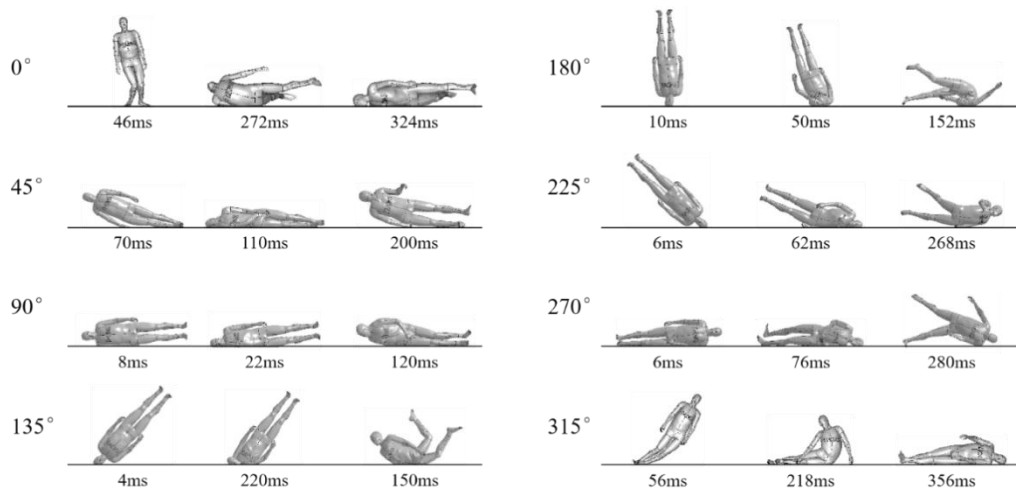

**Figure 5.** Typical landing kinematics in PFOy+.

According to SE, the steps of social engineering optimizer are established according to the above process.

### 3.2. Head Injury Caused by Ground Impact

Figure 6 shows the distribution of HIC values for the head-to-ground contact simulations. It can be seen that the HIC caused by PRA 0–135° is significantly higher than that caused by PRA 180–315°, the result is consistent with the findings of [21]. In addition, there are significant differences in head HIC values caused by different PFOs even under the same PRA. For example, at PRAs 0°, 45°, 90°, and 135°, the HIC values caused by PFOy+ are much lower than those of the other two PFOs. While at PRA 180° and 225°, there was no significant difference in PFO-induced HIC values.

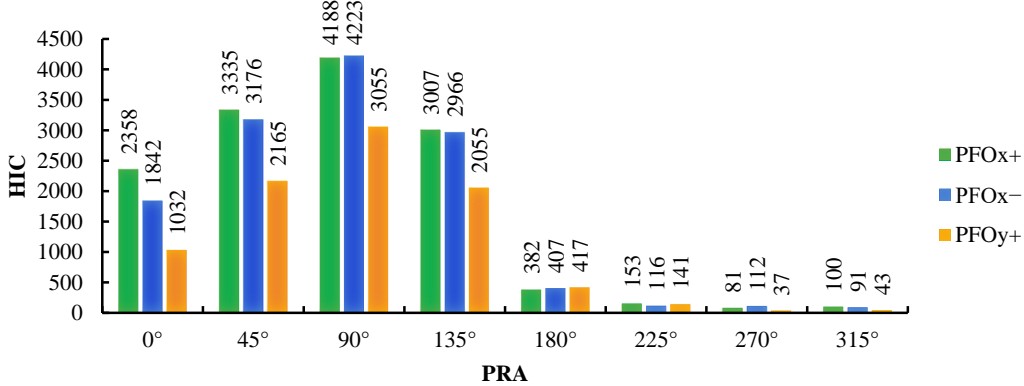

**Figure 6.** HIC values caused during head-to-ground contact.

### 3.3. Neck Injury Caused by Ground Impact

Figure 7 shows the maximum rotation angle of T1 relative to the head. For PFOx+ and PFOx−, the rotation direction of T1 relative to the head is opposite no matter which PRA the pedestrian lands with. When the PRA is in the range of 0–180°, PFOx+ subjects the neck to buckling loads, while PFOx− subjects the neck to extension loads. Conversely, when the PRA is in the range of 225–315°, PFOx+ subjects the neck to extension loads, while PFOx− subjects the neck to buckling loads. It is important to note that the risk of neck injury is greatest in each PFO when the PRA is 180°, followed by 315° and 135°. For PFOy+, it primarily subjects the neck to lateral flexion loads. For PRA 180°, the risk of neck lateral flexion injury was greatest, followed by 135°. When the PRA is approximately 45° or 270° (i.e., the body is more parallel to the ground), the risk of neck injury from the ground contact is almost minimal regardless of which PFO the pedestrian lands.

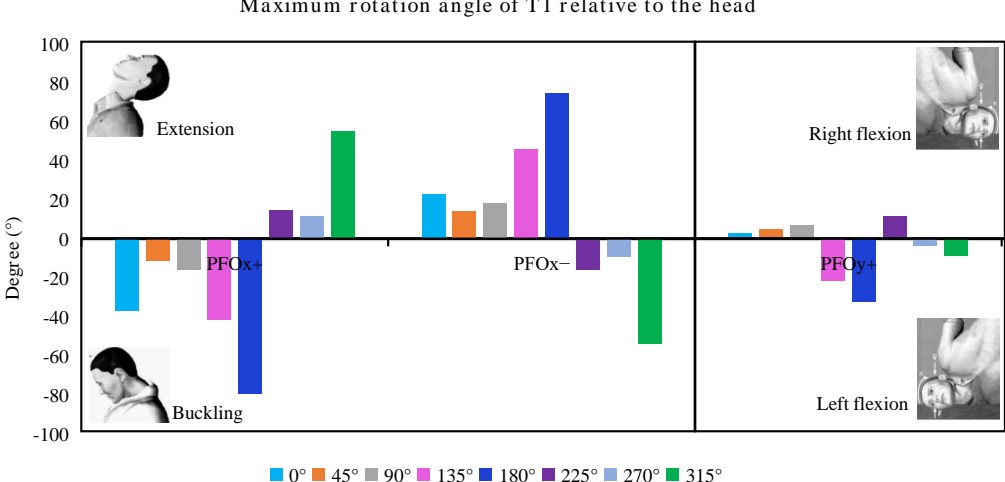

**Figure 7.** Maximum rotation angle of T1 relative to the head.

## 4. Discussions of Results

Combined with pedestrian landing kinematics, the biomechanical mechanisms of pedestrian injury under different ground impact conditions is deeply analyzed as follows.

- *For PRA 0°*

When pedestrians land in the orientation of PFOx+, the feet contact the ground first. Due to the counterclockwise rotational momentum of the body, the pedestrian's knees will bend forward until kneeling. Meanwhile, the upper body leaned backward and eventually the back of the head collided with the ground. In addition to serious head injuries, this landing position would also lead to the risk of waist injury due to excessive reverse stretching of the tissues and organs.

When pedestrians land in the orientation of PFOx−, the feet also contact the ground first. Different from the PFOx+ condition, the change of pedestrian kinematics is not only affected by the counterclockwise rotational momentum, but also the front-to-back asymmetry of the body anatomical structure. Restricted by the physiological bending direction of the knee joint, the impact energy (i.e., rotational momentum energy and landing potential energy) suffered by the lower limbs in this landing orientation is much larger than that in the PFOx+. This can easily lead to reverse traction injuries of the knee joint, and also subject the meniscus, tibia, and femur to greater longitudinal impact. Subsequently, the upper body rotates counterclockwise under the residual rotational energy until the head collides with the ground without any body part protection. In conclusion, this landing orientation is prone to serious head injuries, strained knee ligaments, ruptured menisci, and even fractures of the tibia and femur.

When pedestrians land in the orientation of PFOy+, the right lower limb becomes the main weight-bearing region with the load of the impact energy (i.e., rotational momentum energy and landing potential energy). It would cause the risk of right knee joint medial collateral ligament strain and tibia fracture. Eventually, the right side of the pedestrian's upper body collides with the ground, which could cause injury to the right shoulder and chest and abdomen organs, but the posture could supply a certain buffer effect on the head-to-ground contact.

- *For PRAs 45°, 90°, 225°, and 270°*

When pedestrians land at these four PRAs, the torso and lower limbs remained relatively straight during the ground contact, indicating that the impact energy was less absorbed by these two parts. As for the head, when pedestrians land in orientation of PFOx+ and PFOx−, the head collides with the ground directly; that is, the torso provided no protection on the head. However, when pedestrian lands in the orientation of PFOy+, the shoulder could protect the head to a certain extent, which is why the HIC values in orientation PFOy+ are lower than the other two orientations (see Figure 6).

- *For PRAs 135° and 180°*

Although the head is the first ground impact region for these two PRAs, the HIC values show significant differences. For PRA 135°, the head has—in addition to its linear velocity component perpendicular to the ground—an angular velocity component that rotates toward the ground. Whereas for 180°, the head angular velocity component is parallel to the ground. Therefore, pedestrian landing at PRA 135° would result in a higher HIC value than PRA 180°. In addition, when pedestrians land with PRA 135° in PFOy+, the shoulder has a certain protective effect on the head, resulting in a lower HIC value than the other two orientations.

It should be noted that no matter which PFO the pedestrian lands in, the neck bending caused by these two PRAs would be very severe. Especially PRA 180°, although the HIC value of the head is low, the risk of injury is very high due to the extreme bending of the neck caused by the downward pressure of the body.

- *For PRA 315°*

Figure 6 shows that PRA 315° is friendly to the head-to-ground contact injury risk. However, the kinematic performance (Figures 3–5) suggest that the PRA 315° may cause serious injuries to the pedestrian's chest, neck, and abdomen. In Figure 3, PFOx+ results in over-extension of the neck, increasing the risk of whiplash injury. In Figure 4, PFOx− results in hyper buckling of the neck, along with excessive compression of the chest and abdomen, increasing the injury risk to the internal organs. In Figure 5, PFOy+ causes the lateral hip to strike the ground, potentially resulting in pelvic injury and crush injury to abdominal organs.

**5. Conclusions, Limitations, and Future Works**

This study demonstrated that pedestrian landing kinematics and injuries are significantly influenced by the symmetrical/asymmetrical features of human anatomy. PRA mainly works by changing the sequence of contact sites between the body and the ground. PFO mainly works by changing the specific location of the contact site between the body and the ground.

The head HIC value caused by PRA 0–135° was much higher than that of PRA 180–315°. The head HIC values caused by PFOx+ and PFOx− were not significantly different, but significantly higher than that of PFOy+. That is, PFOy+ is a safer landing orientation compared with PFOx+ and PFOx−.

For the neck, both PRA and PFO shows significant effect on neck injury risk. For PRA 180°, the pedestrian would suffer the greatest neck injury no matter which PFO they land in. For PRA 90 or 270° (that is, the body falls parallel to the ground), the risk of neck injury caused by ground is relatively low. In general, the risk of neck injury caused by PFOy+ is significantly lower than that caused by PFOx+ and PFOx−, indicating PFOy+ is a friendly landing orientation for neck.

The results show that PRA and PFO can be used to analyze landing injury risk. In the follow-up, it should be combined with the artificial intelligence methods [35,36] to jointly provide theoretical support for the formulation of active and passive security protection strategies. This would help to predict the risk of urban traffic accidents and make optimal decisions in future smart cities.

Although this study reported a significant contribution for the evaluation of urban traffic accidents, there are many limitations which can be considered for our future works. First of all, there is a limited number of reconstructed pedestrian accident cases, this results in a lack of statistical significance for the average velocity components in representing the boundaries of pedestrian landing. Therefore, in future research, we need to collect more real-world pedestrian accident data to optimize this research result. In addition, the pedestrian model currently involved in accident reconstruction lacks data validation from ground impact experiments. Therefore, this study only focuses on the distribution trend of pedestrian injury values, rather than the absolute value. Last but not least, in 2020, the COVID-19 pandemic increased the urban traffic as most of people prefer to use their cars instead of public transport [37]. In this regard, the consideration of urban traffic before and during the COVID-19 pandemic is another future research recommendation. Finally, a combination of our prediction model with novel heuristics, metaheuristics, and exact reformulations can be considered for our future work [38–41].

**Author Contributions:** Methodology, L.S., M.L. and Q.Z.; Software, M.L. and Q.Z.; Validation, Y.L., K.C. and H.Z.; Formal analysis, H.Z.; Writing—review and editing, H.Z., A.M.F.-F. and Y.L. All authors have read and agreed to the published version of the manuscript.

**Funding:** This research was funded by the project "China Mobile Traffic Accident Scenario (CTAS)" of CAERI Automotive Safety Technology Center (no. CPYF202004-GA-001); the National Natural Science Foundation of China (no. 52105523); the Natural Science Foundation of Shandong (no. ZR2021QE249) and the China Postdoctoral Science Foundation Funded Project (no. 2021M703559).

**Institutional Review Board Statement:** Not applicable.

**Informed Consent Statement:** Not applicable.

**Data Availability Statement:** The study did not report any data.

**Acknowledgments:** The authors would like to thank Na Du (southern campus of the second hospital of Shandong University) for the validation assistance.

**Conflicts of Interest:** The authors declare no conflict of interest.

## Appendix A

See Figure A1.

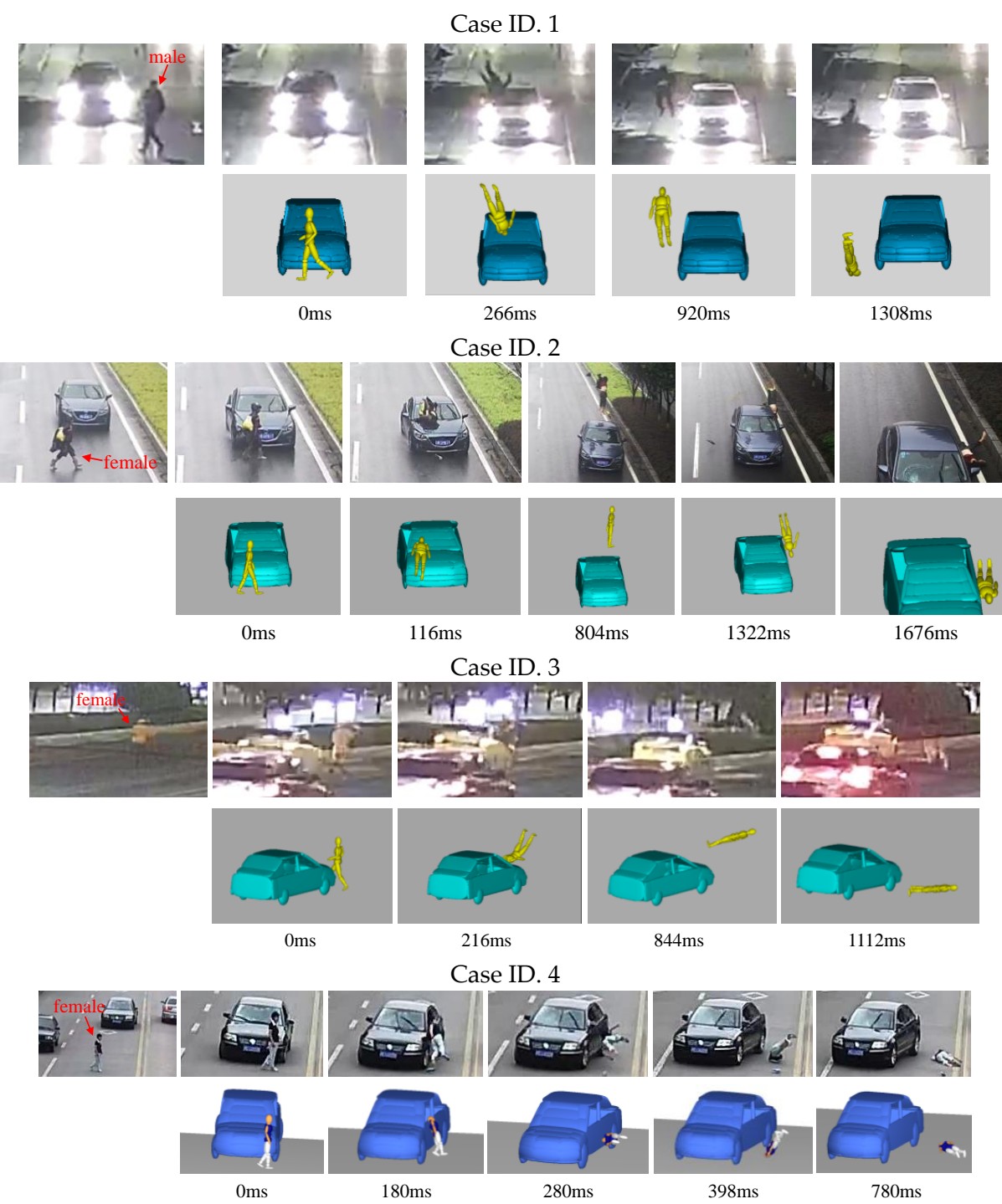

**Figure A1.** *Cont.*

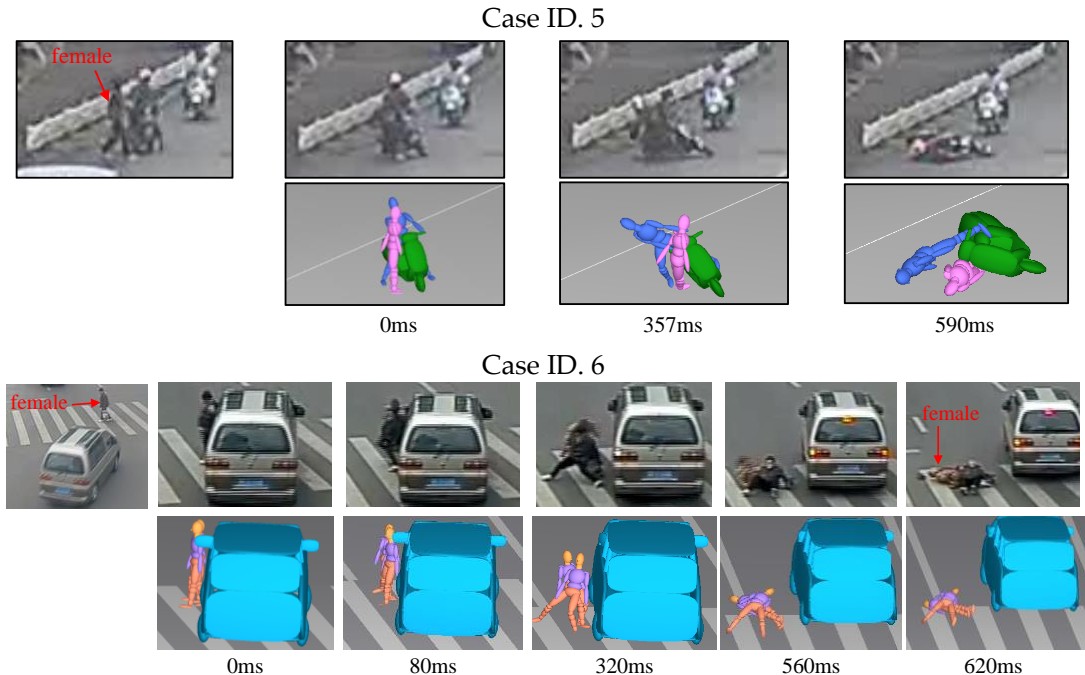

**Figure A1.** Comparison of reconstructed kinematics with video screenshots. The pedestrian's body is blurred in the screenshot, but the body can be clearly judged in the video.

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
