# Peer review of "Evaluation of Urban Traffic Accidents Based on Pedestrian Landing Injury Risks"

_applsci, doi:10.3390/app12126040_

Round 1

Reviewer 1 Report

This paper addresses an interesting topic of risk analysis of pedestrian landing injury based on urban traffic accidents in smart cities. Despite the merits, there are some issues to consider before the paper's publication, mainly the connection to smart cities.  

Introduction

Please explain the concept of smart cities and how this concept is linked with your research.

Methodology

Seems ok to me

Results

Seems ok to me

Discussion

It would be advisable to indicate what limitation has been recorded in the research. Also, further research directions must be mentioned and completed on how your results will be used in the smart cities concept.

Conclusions

Seems ok to me

Overall is article good and I wish the authors good luck in further research.

Author Response

-Reviewer 1

This paper addresses an interesting topic of risk analysis of pedestrian landing injury based on urban traffic accidents in smart cities. Despite the merits, there are some issues to consider before the paper's publication, mainly the connection to smart cities.

Introduction

Please explain the concept of smart cities and how this concept is linked with your research.

Authors’ response: Thank you very much for your comments. The authors inquired the definition of smart city and gave an understanding, as follows:

  • A smart city is an urban area that uses different types of electronic data collection sensors to supply information which is used to manage assets and resources efficiently. - Wikipedia
  • A smart city is a municipality that uses information and communication technologies to increase operational efficiency, share information with the public and improve both the quality of government services and citizen welfare. - IoT World
  • A smart city is a developed urban area that creates sustainable economic development and high quality of life by excelling in multiple key areas; economy, mobility, environment, people, living, and government. - Business Dictionary
  • "Hundreds of aging cities have embraced digital technology, but few are moving as quickly as New York to link municipal computer networks, develop novel applications, make digital data public or install so many thousands of sensors to monitor urban life—from water quality, traffic and power use, to the sound of gunfire." - Robert Lee Hotz, As The World Crowds In, Cities Become Digital Laboratories, Wall Street Journal

For the author's point of view, smart city gathers data by way of connected technology that can be analyzed in order to better inform decisions. As the most important part of the city, the safety of human life is the most fundamental need. Every year, the number of casualties caused by traffic accidents is huge, causing serious losses to families and society, and bringing challenges to the construction of smart cities. In-depth exploration of the mechanism of traffic accidents, formulating safety strategies from the aspects of people, vehicles, roads, and the environment, and providing the best decision-making for people's safety, is a concrete manifestation of building a smart city.

Methodology

Seems ok to me

Results

Seems ok to me

Discussion

It would be advisable to indicate what limitation has been recorded in the research. Also, further research directions must be mentioned and completed on how your results will be used in the smart cities concept.

Authors’ response: Thank you very much for your comments, the authors added the section of Limitation before Conclusion, as follows:

The main limitation for the current study is that there is a limited number of reconstructed pedestrian accident cases, this results in a lack of statistical significance for the average velocity components in representing the boundaries of pedestrian landing. Therefore, in future research, we need to collect more real-world pedestrian accident data to optimize this research result. In addition, the pedestrian model currently involved in accident reconstruction lacks data validation from ground impact experiments. Therefore, this study only focuses on the distribution trend of pedestrian injury values, rather than the absolute value.” (See the blue text in Limitation).

The authors added the future perspective in Conclusion, as follows:

The results show that PRA and PFO can be used to analyze landing injury risk. In the follow-up, it should be combined with the artificial intelligence methods [35, 36] to jointly provide theoretical support for the formulation of active and passive security protection strategies. This would help predict the risk of urban traffic accidents and make optimal decisions in future smart cities.” (See the blue text in Conclusion).

Conclusions

Seems ok to me

Overall is article good and I wish the authors good luck in further research.

Authors’ response: Thank you very much for your comments.

Reviewer 2 Report

1. The state of the art is reduced to two paragraphs at the end of section 1 (introduction) with no detailed discussion and criticism.

2. Please add a paragraph for introducing the structure of the rest of the paper (at the end of section 1)

3. The modeling of the problem needs more enhancement to be more clear. It seems to be a part of the simulation scenarios.

4. A small number (24) of pedestrians were used. For more accuracy of the results, it is highly recommended to increase the number of pedestrians involved in the experiments. 

5. Please add future directions/perspectives of the study at the end of the conclusion. As an example of perspective, I suggest to indicate the interest of applying intelligent AI methods for the analysis of urban traffic accidents risks. Here some references highlighting the interest of AI methods for complex real-world problems such yours: https://doi.org/10.1109/IACS.2019.8809127 and https://doi.org/10.3390/info9110284

Author Response

Dear reviewer

We would like to thank you for the comprehensive and insightful review of our paper. We modified the paper based on the comments. The responses to your comments (designated with the blue color text) are found below.

  1. The state of the art is reduced to two paragraphs at the end of section 1 (introduction) with no detailed discussion and criticism.

Authors’ response: Thank you very much for your comments, the authors added detailed discussion and criticism in this section, as follows:

In recent years, some researchers have studied the kinematics of pedestrian landing and have come up with several typical types of pedestrian landing kinematics [17, 19, 25, 29], this provided an understanding for pedestrian-to-ground contact, however, they did not deeply analyze the influence mechanism of landing kinematics on landing injuries. In 2018, Shi et al. [21] proposed the concept of the pedestrian rotation angle ranges (PRARs) to illustrated the relationship between pedestrian landing kinematics and landing injuries, which demonstrated that any landing kinematics could be classified as one of the 4 PRARs, and the PRARs are highly correlated with the ground contact mechanism and head injury risk. However, pedestrian landing kinematics and injury risk are not only affected by pedestrian rotation angle, but also by a variety of other factors, such as the pedestrian facing orientations and the landing sequence of body parts, which they did not analyze. Further study found that for the same pedestrian rotation angle, there were significant differences in the risk of head injury during landing for different pedestrian facing orientations [30]. However, they did not delve into the mechanism between this factor and pedestrian injury risk. ” (See the blue text in the third paragraph of Introduction).

  1. Please add a paragraph for introducing the structure of the rest of the paper (at the end of section 1)

Authors’ response: Thank you very much for your comments. The authors revised the content of this section, as follows:

Through in-depth research, it was found that the biomechanical mechanisms of pedestrian injury is closely related to the symmetric/asymmetric phenomena (structures, shapes, morphologies, geometry, direction, models, aesthetics, etc.) in life sciences [31-32]. In response to this intriguing problem, this paper systematically and quantitatively investigated the effect of human anatomy features on body landing injury risk with considering two key kinematic parameters, pedestrian rotation angle (PRA) and pedestrian facing orientation (PFO). To this end, six real-world pedestrian accidents were reconstructed to derive the boundary parameters at the moment of body-to-ground impact. Then, a total of 24 pedestrian landing simulations were carried out using the THUMS v4.0.2 model with considering 8 PRAs and 3 PFOs. The kinematic-based injury criteria such as the Head Injury Criterion (HIC) and the maximum rotation angle of T1 relative to the head were included in the analysis to investigate head and neck injury risk. Moreover, the influence mechanism of human anatomical symmetry/asymmetry on the risk of landing injuries was deeply analyzed. The results of this study are beneficial to provide theoretical guidance for the research on pedestrian injury control strategies in the automotive industry and traffic management departments.” (See the blue text in the fourth paragraph of Introduction).

  1. The modeling of the problem needs more enhancement to be more clear. It seems to be a part of the simulation scenarios.

Authors’ response: Thank you very much for your comments. The authors add the workflow of accident reconstruction in section 2.1 (Parametric Research Proposal), as follows:

The detailed workflow of accident reconstruction is depicted in Figure 2.” (See the blue text in section 2.1).

Figure 2 Workflow implemented for the pedestrian accident reconstruction.

  1. A small number (24) of pedestrians were used. For more accuracy of the results, it is highly recommended to increase the number of pedestrians involved in the experiments. 

Authors’ response: Thank you very much for your comments, the authors added the section of Limitation to illustrate the problem, as follows:

The main limitation for the current study is that there is a limited number of reconstructed pedestrian accident cases, this results in a lack of statistical significance for the average velocity components in representing the boundaries of pedestrian landing. Therefore, in future research, we need to collect more real-world pedestrian accident data to optimize this research result.(See the blue text in Limitation).

  1. Please add future directions/perspectives of the study at the end of the conclusion. As an example of perspective, I suggest to indicate the interest of applying intelligent AI methods for the analysis of urban traffic accidents risks. Here some references highlighting the interest of AI methods for complex real-world problems such yours: https://doi.org/10.1109/IACS.2019.8809127 and https://doi.org/10.3390/info9110284

Authors’ response: Thank you very much for your comments, the authors added the future perspective in Conclusion, as follows:

The results show that PRA and PFO can be used to analyze landing injury risk. In the follow-up, it should be combined with the artificial intelligence methods [35, 36] to jointly provide theoretical support for the formulation of active and passive security protection strategies. This would help predict the risk of urban traffic accidents and make optimal decisions in future smart cities. (See the blue text in Conclusion).

35.Hassanat, A. . Furthest-Pair-Based Decision Trees: Experimental Results on Big Data Classification. Information (Switzerland), 2018, 9.11.

  1. Tarawneh, A. S. ,  Chetverikov, D. ,  Hassanat, A. B. , &  Rahman, M. S. . Deep Face Image Retrieval: a Comparative Study with Dictionary Learning. IEEE 10th International Conference on Information and Communication Systems IEEE, 2019.(See the blue text in Reference).

Reviewer 3 Report

The paper aims to analyze pedestrian accidents in urban traffic, but the authors present only the results obtained after 24 simulations of pedestrian-ground impact using THUMS v4.0.2 software.

From the point of view of  the simulation it is OK, but the results obtained should be compared with real results for a better validation.

The data regarding the 6 accidents that the authors reconstruct in the software product are few. The following should be added: location, type of intersection, road infrastructure, factors related to the accident and only then reconstituted and analyzed with the software.

It is known that the dynamics of a vehicle-pedestrian accident depends on many factors: vehicle speed, pedestrian speed, vehicle braking, pedestrian design distance, etc. and that, shis dynamic involves the following three distinct phases: the contact phase, the air flight phase, the sliding phase.

These are found in the simulation software, but should be specified for the 6 accidents taken into account ( for example: a table).

Conclusions may be developed.

Author Response

Dear reviewer

We would like to thank you for the comprehensive and insightful review of our paper. We modified the paper based on the comments. The responses to your comments (designated with the blue color text) are found below.

1、The paper aims to analyze pedestrian accidents in urban traffic, but the authors present only the results obtained after 24 simulations of pedestrian-ground impact using THUMS v4.0.2 software.

Authors’ response: Thanks for your insightful comments. Since real-world accidents are hard to obtain, we only investigated a small number of valid cases, and this paper selects 6 pedestrian cases for analysis. On this basis, we use the crash simulation software to complete the modeling of the simulation matrix, and then carry out parametric research. This flaw is illustrated in the new section of Limitations, as follows:

The main limitation for the current study is that there is a limited number of reconstructed pedestrian accident cases, this results in a lack of statistical significance for the average velocity components in representing the boundaries of pedestrian landing. Therefore, in future research, we need to collect more real-world pedestrian accident data to optimize this research result. (See the blue text in Limitation).

2、From the point of view of the simulation it is OK, but the results obtained should be compared with real results for a better validation.

Authors’ response: Thank you for your comments. The authors add more detailed validation of reconstruction results, such as kinematics (see figure A1 in Appendix) and final positions (see Table 2), as follows:

Appendix

Case ID. 1 

Case ID. 2 

Case ID. 3 

Case ID. 4 

Case ID. 5 

Case ID. 6

Figure A1 Comparison of reconstructed kinematics with video screenshots. (The pedestrian's body is blurred in the screenshot, but the body can be clearly judged in the video.)

Table 2. Validation and extraction of reconstruction results.

Case ID

Final position error %: reconstructed vs actual.

Linear and angular velocity components (2ms before body-to-ground contact)

Pedestrian

vehicle

Vx(m/s)

Vy(m/s)

Vz(m/s)

ωy(rad/s)

1

3.1%

2.4%

2.8

-0.4

-5.8

4.6

2

3.6%

2.7%

4.7

0.1

-6.9

5.8

3

2.4%

1.7%

3.9

0.2

-6.1

5.2

4

0.9%

0.6%

1.3

-3.8

-5.5

6.5

5

1.2%

1.1%

0.9

-0.8

-5.2

3.8

6

0.6%

1.3%

0.6

-1.4

-4.3

2.4

3、The data regarding the 6 accidents that the authors reconstruct in the software product are few. The following should be added: location, type of intersection, road infrastructure, factors related to the accident and only then reconstituted and analyzed with the software.

Authors’ response: Thank you for your comments. The authors provide more accident-related information, as shown in Table 1:

Table 1. Basic information about the accidents.

Case ID

Vehicle info

Pedestrian info

Road and environment info

Type

Velocity (km/h)

Brake before collision

Gender

Age

Height (cm)

State

Location

Obstacle

Pavement

Humidity

1

Sedan

67

No

Male

36

179

Running

Motor lane

None

Asphalt

Wet

2

Sedan

90

No

Female

51

162

Running

Motor lane

None

Asphalt

Wet

3

Sedan

63

Yes

Female

37

164

Running

Motor lane

None

Asphalt

Dry

4

Sedan

29

No

Female

73

142

Walking

Motor lane

None

Asphalt

Dry

5

E-bike

31

Yes

Female

63

157

Walking

Non-motor lane

None

Asphalt

Dry

6

MPV

29

No

Female

79

158

Walking

Crosswalk

None

Asphalt

Dry

4、It is known that the dynamics of a vehicle-pedestrian accident depends on many factors: vehicle speed, pedestrian speed, vehicle braking, pedestrian design distance, etc. and that, shis dynamic involves the following three distinct phases: the contact phase, the air flight phase, the sliding phase.

These are found in the simulation software, but should be specified for the 6 accidents taken into account ( for example: a table).

Authors’ response: Thank you for your comments. The authors illustrate the 4 phases of a vehicle-pedestrian accident and rework Figure 1. In addition, figure A1 (in Appendix) presents the reconstruction results for the various phases of each accident. Figure 1 is shown as:

Figure 1 Methodology implemented for the parametric research. ”

5、Conclusions may be developed.

Authors’ response: Thank you for your comments. The authors add the future perspective in Conclusion, as follows:

5. Conclusions, limitations and future works

This study demonstrated that pedestrian landing kinematics and injuries are significantly influenced by the symmetrical/asymmetrical features of human anatomy. PRA mainly works by changing the sequence of contact sites between the body and the ground. PFO mainly works by changing the specific location of the contact site between the body and the ground.

The head HIC value caused by PRA 0°~135° was much higher than that of PRA 180°~315°. The head HIC values caused by PFOx+ and PFOx- were not significantly different, but significantly higher than that of PFOy+. That is, PFOy+ is a safer landing orientation compared with PFOx+ and PFOx-.

For the neck, both PRA and PFO shows significant effect on neck injury risk. For PRA 180°, the pedestrian would suffer the greatest neck injury no matter which PFO they land in. For PRA 90 or 270° (that is, the body falls parallel to the ground), the risk of neck injury caused by ground is relatively low. In general, the risk of neck injury caused by PFOy+ is significantly lower than that caused by PFOx+ and PFOx-, indicating PFOy+ is a friendly landing orientation for neck.

The results show that PRA and PFO can be used to analyze landing injury risk. In the follow-up, it should be combined with the artificial intelligence methods [35, 36] to jointly provide theoretical support for the formulation of active and passive security protection strategies. This would help predict the risk of urban traffic accidents and make optimal decisions in future smart cities.

Although this study reported a significant contribution for the evaluation of urban traffic accidents, there are many limitations which can be considered for our future works. First of all, there is a limited number of reconstructed pedestrian accident cases, this results in a lack of statistical significance for the average velocity components in representing the boundaries of pedestrian landing. Therefore, in future research, we need to collect more real-world pedestrian accident data to optimize this research result. In addition, the pedestrian model currently involved in accident reconstruction lacks data validation from ground impact experiments. Therefore, this study only focuses on the distribution trend of pedestrian injury values, rather than the absolute value. At last but not least, in 2020, the COVID-19 pandemic has increased the urban traffic as most of people prefer to use their cars instead of public transports [37]. In this regard, the consideration of urban traffics before and during the COVID-19 pandemic is another future research recommendation. Finally, combination of our prediction model with novel heuristics, metaheuristics and exact reformulations can be considered for our future works [38-41]. ” (See the blue text in Conclusion).

Round 2

Reviewer 3 Report

The authors made all the requested changes.